# Predicting COVID-19 in very large countries: The case of Brazil

**V. C. Parro**[1]☺*, **M. L. M. Lafetá**[1]☺, **F. Pait**[2], **F. B. Ipólito**[1], **T. N. Toporcov**[3]

**1** Instituto Mauá de Tecnologia, Electrical Engineering, São Caetano do Sul, Brazil, **2** Escola Politécnica da Universidade de São Paulo, São Paulo, Brazil, **3** Faculdade de Saúde Pública da Universidade de São Paulo, São Paulo, Brazil

☺ These authors contributed equally to this work.
\* vparro@maua.br

**Data Availability Statement:** The dataset and documentation for this study are available on Kaggle. Dataset: https://www.kaggle.com/ marcelolafeta/epidemicmodelsbrazilstateanalysis. Documentation: https://www.kaggle.com/

## Abstract

This work presents a practical proposal for estimating health system utilization for COVID-19 cases. The novel methodology developed is based on the dynamic model known as **S**usceptible, **I**nfected, **R**emoved and **D**ead (SIRD). The model was modified to focus on the healthcare system dynamics, rather than modeling all cases of the disease. It was tuned using data available for each Brazilian state and updated with daily figures. A figure of merit that assesses the quality of the model fit to the data was defined and used to optimize the free parameters. The parameters of an epidemiological model for the whole of Brazil, comprising a linear combination of the models for each state, were estimated considering the data available for the 26 Brazilian states. The model was validated, and strong adherence was demonstrated in most cases.

## Introduction

In December 2019, the new coronavirus severe acute respiratory syndrome coronavirus 2 (SARS-Cov-2) was first identified in Wuhan, China. On March 11, the World Health Organization designated COVID-19 as a pandemic. As of August 2020, more than 23 million COVID-19 cases and 80,000 deaths had been reported worldwide [1]. In Brazil, the virus was first identified in São Paulo city on February 26, and the first death occurred in March in Rio de Janeiro. The new cases detected at the beginning of the pandemic largely coincided with Brazilian cities with airport access, with approximately 2 million Brazilians exposed in approximately 20 weeks. As of the 7th week after the first case, the virus had spread to cities without airports, probably via road transport, increasing the population at risk. Within 5 weeks, according to Wesley Cotta/Ministry of Health data [2], all Brazilian states had registered active cases of the disease. In addition to the different characteristics of the states, which have HDI index values ranging from 0.631 for the state of Alagoas to 0.824 for the state of Distrito Federal, different measures were taken to achieve social isolation, implying different courses of the pandemic.

Brazil, with a population of approximately 200 million inhabitants, is composed of 26 states and the federal district. Approximately 30 years ago, the country implemented a universal and

marcelolafeta/covid-brazil-local-predictions-heuristic-learning.

**Funding:** This work is supported by Instituto Mauá de Tecnologia.

**Competing interests:** The authors have declared that no competing interests exist.

decentralized health system (Sistema Único de Saúde) [3]. There was instability in the federal management of the health crisis caused by the pandemic, with a number of changes in the Ministry of Health during a short period of time. States independently made several important decisions for controlling the epidemic, leading to high heterogeneity in the non-pharmacological measures taken to mitigate the pandemic. The utilization of the health system and decisions about isolation guidelines served as a guide for most of the official communications from states (26) and municipalities (5570) in the Brazilian press [4].

Internationally, machine learning has been widely used to predict disease behaviour, to forecast demand for health services, to plan and evaluate measures to reopen quarantined sites [5–7], and also for medical diagnoses [8]. The choice of the best model to forecast demand has been discussed in the literature and remains controversial. COVID-19 is a new disease, and its transmission dynamics and natural history are not yet completely clear; in addition, there is a variable proportion of asymptomatic and mildly symptomatic cases. Those are not notified to health authorities, and although individuals affected with milder cases do not need treatment, they may transmit the disease [9]. The modeling challenge is even greater in large countries with significant inequality and a heterogenous evolution of the pandemic, such as Brazil. This country has high-quality data on COVID-19, which originate from the epidemiological surveillance of acute influenza and respiratory syndromes, and are available electronically.

We propose a modification of the Susceptible, Infected, Removed, and Dead (SIRD) [10–13] model to describe the dynamics of health system usage based on reported cases only, and not on the epidemic as a whole [14]. A comparison between models applicable to CoViD19 can be found in [15], where the authors test eight empirical functions, four methods of statistical inference, and five dynamic models built from variations of the SIR model, all of them with data from the epidemic in China. In their work, the models are compared using the Akaike information criterion (AIC), mean square error (MSE) and robustness index, allowing assessment of overestimation and underestimation. In the specific case of dynamic models, they establish a cost-benefit relationship between model complexity and predictive capacity.

The simplicity of the SIRD model compared to more complete and sophisticated models [16, 17] makes it easier to tune its parameters and simplifies its use by public agents in management and public communication. Only a portion of those infected by COVID-19 will use the health system. It is known that the peak of the SIRD model is dependent on the population considered. This led us to consider a weighting of the total population to estimate the utilization of the public health system. The proposal was validated by applying the model to each Brazilian state. Tuning the modified SIRD model for each state permits a comparative assessment of its main parameters, infection rate and removal rate, in addition to an assessment of the basic reproduction rate $R_0$ and the effective reproduction rate $R_t$, all of them relevant parameters for public health management and decision making [18]. The model can be used together with solutions for tracking individuals with the purpose of monitoring the epidemic in contagion and geographic location [19].

The global model for all of Brazil was obtained from a linear combination of the estimated active cases for each state. The main contribution of the novel model developed is the demonstration that the data from Brazil as a whole does not follow a simple SIRD model, and the prediction that the epidemic would intensify in the second half of the year due to the natural risk associated with its presence in different states and locations. We have proposed an algorithm that describes this behaviour. Additional important contributions are the possibility of using the model estimates to predict the infection rate and the reproducibility index for the whole country. These indices, which can be reliably estimated from our model, are important for public management and can be easily communicated to society.

## The model

In this section, the machine learning problem is introduced based upon the so-called SIRD model. With this particular model structure, an optimization algorithm using a heuristic search is introduced into the learning algorithm. Additionally, a data-driven optimization technique is introduced as a solution to the susceptible data unavailability problem by introducing a degree of freedom to the algorithm. To start the optimization problem structure, consider the SIRD differential model described by equations Eqs (1)–(4):

$$\frac{dS(t)}{dt} = \frac{-\beta I(t)S(t)}{P}, \tag{1}$$

$$\frac{dI(t)}{dt} = \frac{\beta I(t)S(t)}{P} - \gamma I(t) - \mu I(t), \tag{2}$$

$$\frac{dR(t)}{dt} = \gamma I(t), \tag{3}$$

$$\frac{dD(t)}{dt} = \mu I(t), \tag{4}$$

where $\beta$, $\gamma$ and $\mu$ are the average number of contacts per person per period of time, the inverse of the number of days required for a person to pass from the infected to the recovered state, and the average number of deaths per period of time. The continuous time series of the susceptible, infected, removed and death are represented respectively by $S(t)$, $I(t)$, $R(t)$ and $D(t)$. Considering that the existing data sets are usually sampled uniformly, there is an advantage of using the discrete representation of the SIRD model, which can be achieved by Eqs (5)–(8), where $\Delta t$ is the sample time of the data-sets, $P$ is the total population that should be considered, and the discrete time series of the susceptible, infected, removed and death are represented respectively by $S(k)$, $I(k)$, $R(k)$, and $D(k)$.

$$S(k+1) = S(k) + \Delta t(-\beta I(k)S(k)/P), \tag{5}$$

$$I(k+1) = I(k) + \Delta t(\beta I(k)S(k)/P - \gamma I(k) - \mu I(k)), \tag{6}$$

$$R(k+1) = R(k) + \Delta t\gamma I(k), \tag{7}$$

$$D(k+1) = D(k) + \Delta t\mu I(k), \tag{8}$$

From model Eqs (5)–(8), determining the mean squared error $e(k + 1)$ of the model from the provided data is straightforward. Therefore it is possible to consider the error equation with an aggregated value for each component, such as the maximum value of each component, resulting in the weighted error given by Eq (9). The complete set of data for each model variable is represented by the vectors **S**, **I**, **R**, and **D**:

$$e_p(k) = \frac{(S(k) - \tilde{S}(k))^2}{\|\mathbf{S}\|} + \frac{(I(k) - \tilde{I}(k))^2}{\|\mathbf{I}\|} + \frac{(R(k) - \tilde{S}(k))^2}{\|\mathbf{R}\|} + \frac{(D(k) - \tilde{S}(k))^2}{\|\mathbf{D}\|}, \tag{9}$$

where the components with the upper tilde are the output components of the differential model (5)–(8) for a particular set of parameters $\beta$, $\gamma$ and $\mu$, e.g., the component $\tilde{I}(k)$ is the simulation of Eq (6) for a particular set of parameters at $k$ time instants from the initial sample. From that, it is possible to compute the mean squared error using Eq (10), where $N$ is the

number of data samples.

$$\text{MSE} = \frac{\sum_{k=0}^{N} e(k)}{N} \qquad (10)$$

The MSE defined could be employed as the cost function for the data-driven problem, but due to high amplitude difference of the model components mean values, the cost function must take into consideration a weighting parameter. This parameter is used to attribute the same importance to the error of each component of the model. This equalizes the importance of all components on the cost function, and enhances the backward and forward stability of the optimization search space.

Notice that to simulate the components, $\tilde{S}(k), \tilde{I}(k), \tilde{R}(k),$ and $\tilde{D}(k)$, the learning algorithm must solve Eqs (5)–(8) for a particular set of parameters. This is straightforward provided that the initial conditions $I(0)$, $R(0)$, and $D(0)$ are known, as they are related to the size of the population $P$ by

$$P = S(k) + I(k) + R(k) + D(k). \qquad (11)$$

### Definition of susceptible component

The fraction of susceptible individuals is usually not available on data-sets. It is usually computed from Eq (11), using the components $I(k)$, $R(k)$, $D(k)$, and the estimated population size $P$. But this assumption is not quite accurate, as the entire population, $P$, cannot be considered susceptible, specially in case of COVID-19. The model we propose will fit the data-set when the information provided by is actually the number of people who visited a health care facility, and are then tracked by the data. In the next section, we will discuss an algorithm capable of computing the influence of the susceptible component into the cost function.

The susceptible component is subject to imprecision because it depends on environmental aspects such as isolation, disease health impact, targeted people, and even climate conditions. Several studies suggest that it should not be used in the optimization problem. This is problematic: to predict the behavior of the epidemic, it is necessary to consider the initial value of the susceptible components in (11), the value of the considered population size. For example, we could determine the susceptible component value at the time where (2) is zero. So for that can write (2) when $t = t_p$ resulting in

$$\frac{dI(t_p)}{dt} = \frac{\beta S(t_p) I(t_p)}{P} - \gamma I(t_p) - \mu I(t_p) \triangleq 0 \qquad (12)$$

where $t_p$ is the instant of $t$ where the peak occurs. This leads to the condition

$$S(t_p) = \frac{\gamma + \mu}{\beta} P \qquad (13)$$

which shows that the peak moment $t_p$ depends on the correct selection of the population size, $P$. Given a susceptible component value computed from (11) and a population size $P$, the parameters $\beta$, $\mu$ and $\gamma$ in (13) are bounded by this particular representation. Now consider that the correct initial value of the susceptible component, $S(0)$, is not the total population, but actually a proportion of it, $\lambda P$. This happens in scenarios where part of the population is immune or are not impacted by the disease symptoms, and therefore are only carriers. In this

particular case it is possible to rewrite (13) as

$$S(t_p) = \frac{\gamma + \mu}{\beta} \lambda P. \tag{14}$$

Any distortion of the susceptible component, or of the population value, will be acknowledged by the new degree of freedom of the model, $\lambda$. The previous parameters are guided by their particular components, (2)–(4), and the existent data-set. The susceptible component can be computed by considering $\lambda$ from (15). Considering the characteristics of CoViD 19, where not all infectious people seek the health care system, and that our interest is to model the dynamics of the people who use to the health care system, and specially estimating the peak $S(t_p)$, the weighting of the total population $\lambda P$ allows the SIRD model to represent this dynamic:

$$\lambda P = S(k) + I(k) + R(k) + D(k). \tag{15}$$

## Optimization problem

The optimization problem can be structured in the form

$$\arg_{\{\beta, \gamma, \mu, \lambda\} \in \Omega} \min \frac{\sum_{k=0}^{N} e_p(k)}{N} \tag{16}$$

with $e_p(k)$ representing the weighted error at sample $k$, given by (9), and the component data reference of $S(k)$ being obtained by (15). In the usual formulation of heuristic optimization algorithms, the arguments must be bounded by the search space $\Omega$.

The search argument boundaries determination is straightforward as each parameter presents a physical interpretation in the model.

- $\beta$ is the amount of people one contagious individual infects per time unit;

- $\gamma^{-1}$ is the amount of time that an infected individual takes to recover;

- $\mu$ is the proportion of infected people that dies per time unit;

- $\lambda$ is the proportion of the population considered as initially susceptible.

Limits for each of these parameters are given by common sense. Even better is to obtain them numerically, considering the influence of the basic reproduction number, $R_0$. For this model, $R_0$ can be obtained from the relation

$$R_0 = \frac{\beta \lambda}{\gamma}. \tag{17}$$

The basic reproduction number measures the average number of people one contagious person will infect during the contamination period. When $R_0 > 1$, one person will infect more than one other, and therefore the disease will be capable of self-sustaining growth. Conversely, when $R_0 < 1$, the disease by itself will not become epidemic. There is a vast literature concerning the value of $R_0$ for the most common epidemics. We propose the following approach for solving the optimization problem (16): instead of searching for the set of arguments $\{\beta, \gamma, \mu, \lambda\}$, we search for $\{R_0, D, \mu, \lambda\}$, where $D = \gamma^{-1}$. The search for $R_0$ and $D$ is better conditioned then the search for $\beta$ and $\gamma$, as $R_0$ directly define the existence of the epidemic.

The optimization problem can thus be rewritten in the form

$$\arg \min_{\{R_0,D,\mu,\lambda\}\in\bar{\Omega}} \frac{\sum_{k=0}^{N} e_p(k)}{N} \tag{18}$$

where $\bar{\Omega}$ is the new search space, considering $R_0$ and $D$.

## Validation for Brazilian states

Brazil is a large country, and there are several cultural and environmental aspects that make its states diverse. Each state can be treated as an isolated epidemic environment, and we can fit a model for each individual state. Fig 1 shows the data and the model predicted for two distinct Brazilian states, Maranhão and São Paulo. Using all data to fit each model, it is possible to see that in scenarios where the data were collected rigorously and strict isolation was implemented, such as in Maranhão, the algorithm was able to fit the data pattern with high fidelity. Even in scenarios where the data were not collected properly, such as São Paulo, the model was able to visualize the main pattern of the data.

The modified SIRD model exhibited strong adherence to the data for most states with R2 values between 0.99 and 0.82. Fig 1 illustrates the adjustment of the model for the date 7/21/20 for the state of Maranhão, which has a population of approximately 7 million inhabitants (approximately 20 inhabitants per km$^2$) and declared 12 days of lockdown in May 2020; and the state of São Paulo, which has a population of approximately 46 million inhabitants (approximately 166 inhabitants per km$^2$). The basic reproducibility index estimated for Maranhão was $R_0 = 3.52$ and the basic reproducibility index estimated for São Paulo was $R_0 = 4.78$.

In Fig 2, the $R^2$ values for all states are presented for each scenario: searching for $\{R_0, D, \mu, \lambda\}$ and searching for $\{R_0, D, \mu\}$ with the previous setting $\lambda = 1$. For each state, $R^2$ is smaller when $\lambda$ is selected by the model, i.e., the extra degree of freedom is considered in the population size. Without the reduction of the population considered, there is no set of values for the argument $\{R_0, D, \mu\}$ that can properly fit the data set analysed. Note that this fitness performance of the algorithm is only possible due to the new degree of freedom introduced represented by the parameter $\lambda$.

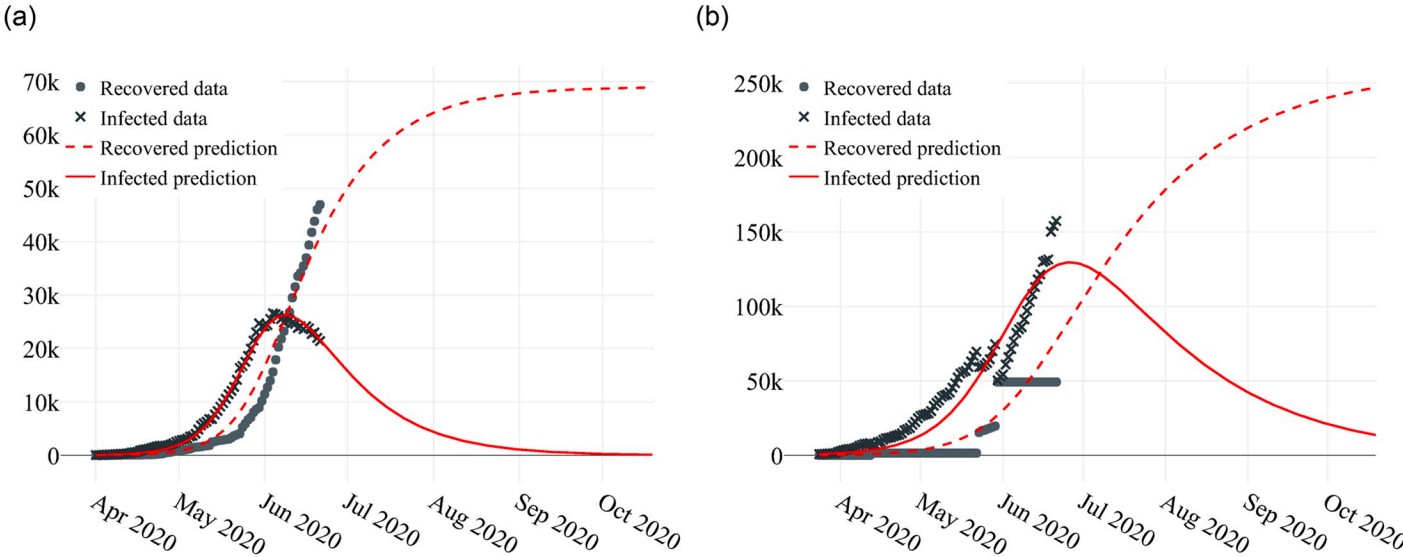

**Fig 1. Comparison of predictions using the estimated models, with each state real data.** (a) Maranhão results. (b) São Paulo results.

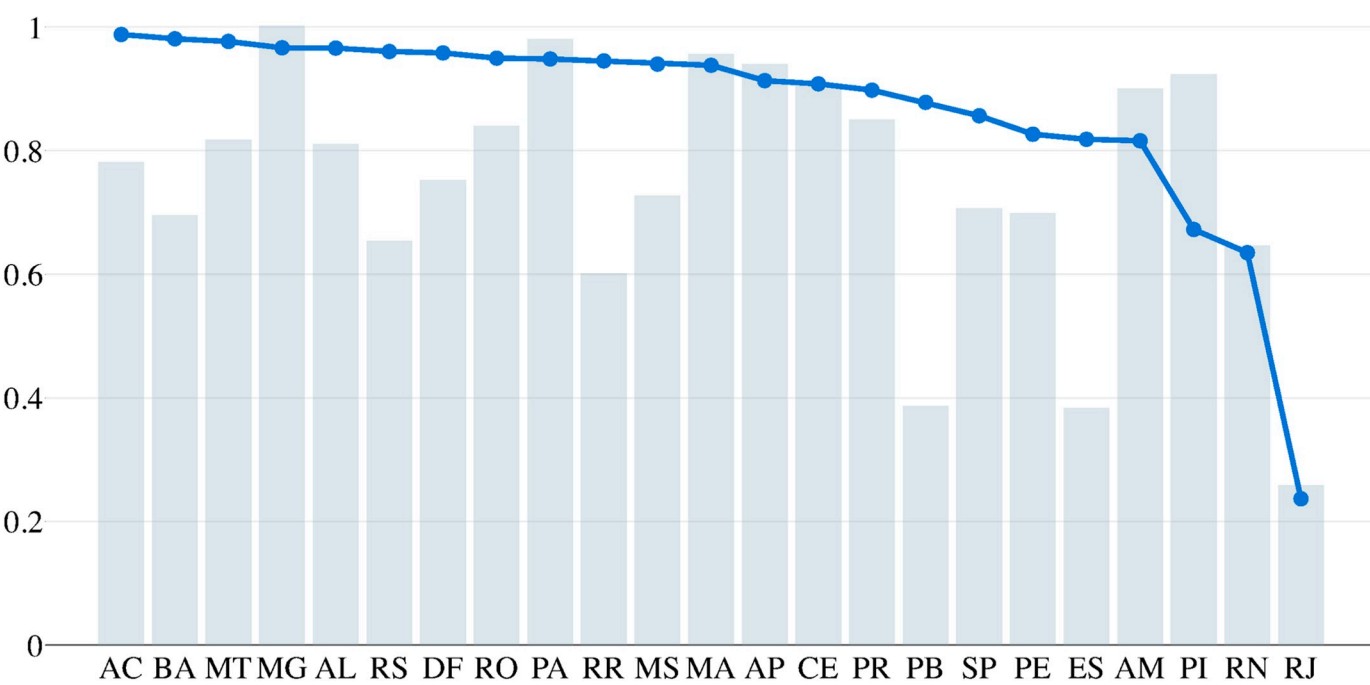

**Fig 2. $R^2$ value comparing the predictions with the real data of each state and the reliability metric that shows the proportion of the data that is reliable to use on the learning algorithm.**

## Results from the model

The global model for Brazil is determined via the linear combination of state models. The epidemic curve of active cases, estimated on June 21, 2020, can be analysed in Fig 3. The peak observed in June 2020 is strongly influenced by the peak in the state of São Paulo. The decay of the curve followed by support indicates that Brazil is expected to have a stable number of active cases until September or October 2020 and an increasing number of active cases until the first quarter of 2021. An important finding is that application of the SIRD model to Brazil as a whole (dotted in the figure) results in a different prediction for the case dynamics, indicating control of the epidemic in October 2020. The simulation was carried out on 06/27/2020 and the result clearly demonstrates that the model for Brazil does not follow the SIR model as well. The proposed model shows more realistic behaviour about the duration of the epidemic. Some models mistakenly predicted the end of the epidemic at the end of August 2020 [20].

In Fig 4, the predicted value for use of the Brazilian health system is presented, based on the average proportion of the population that will attend hospitals and health centres throughout the epidemic. In Fig 4, this value is shown as more data were provided to the model, i.e., as the epidemic progressed in the country. The value becomes close to the current (most recent) value of 1.0% of the population on 5/17/20, approximately 1.5 months before the first peak, when the predicted use was 0.8% of the country's population.

It can also be seen in Fig 4 that the growth of the number of individuals seeking health care and its future predictions have approximately linear behaviour, implying a constant growth rate, which indicates that the number of new cases is stable. This behavior is consistent when

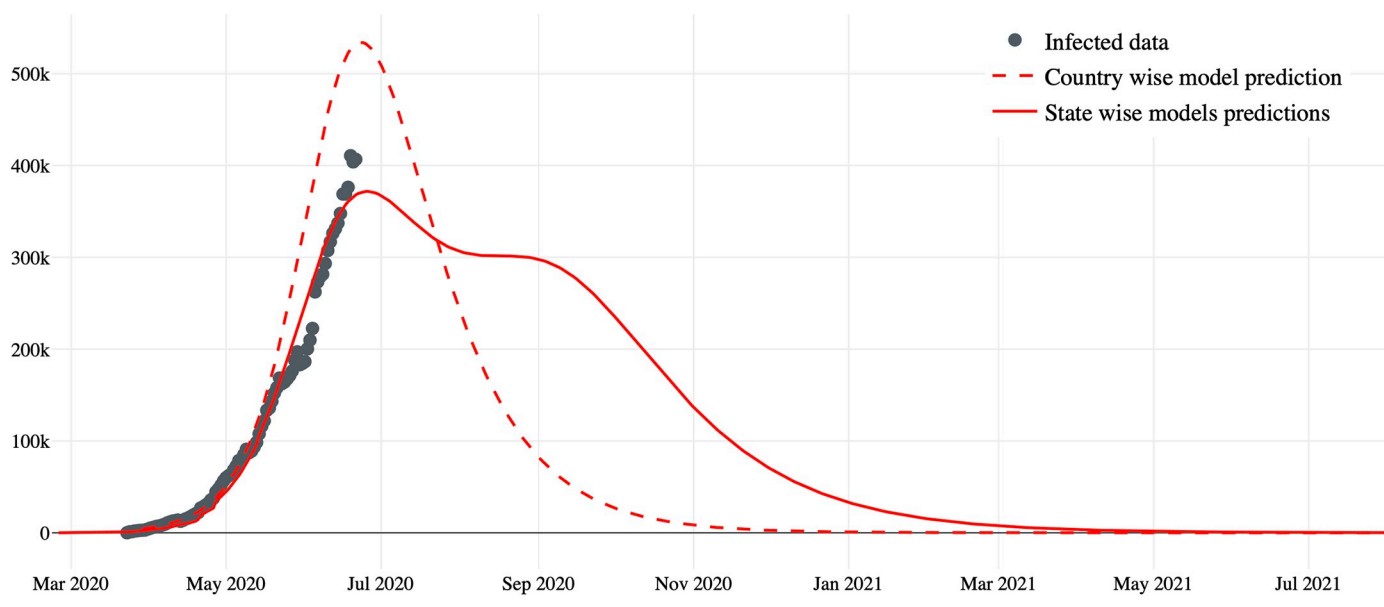

**Fig 3. Comparison of predictions using the estimated models with real data for each state.**

analyzed in the light of new cases and especially the number of deaths, approximately constant, which tends to be more reliable. According to the model, this rate should drop starting in mid-October, as shown in Fig 3.

## Results by state

One of the challenges of public health management was described as "flattening" the epidemic curve, a way to openly communicate the strategies chosen for this purpose. Fig 5 illustrates the peaks in Brazil, with the first state to reach the peak being Pernambuco and the last states to

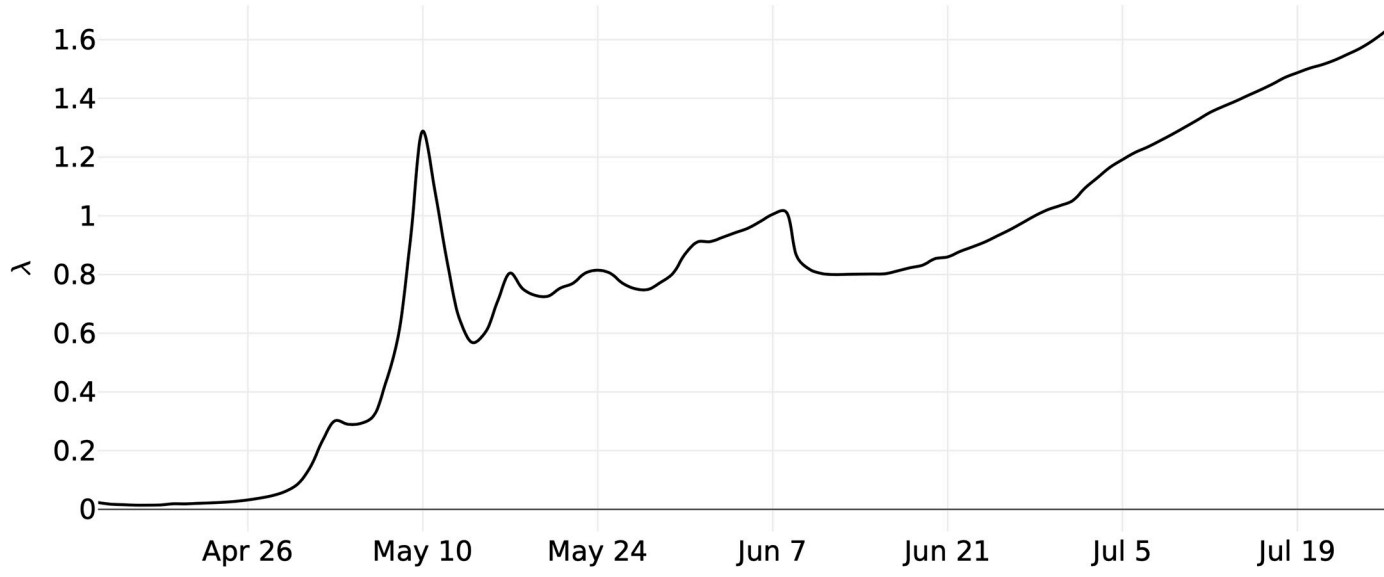

**Fig 4. Prediction of the proportion of the population to attend to health care systems in Brazil.**

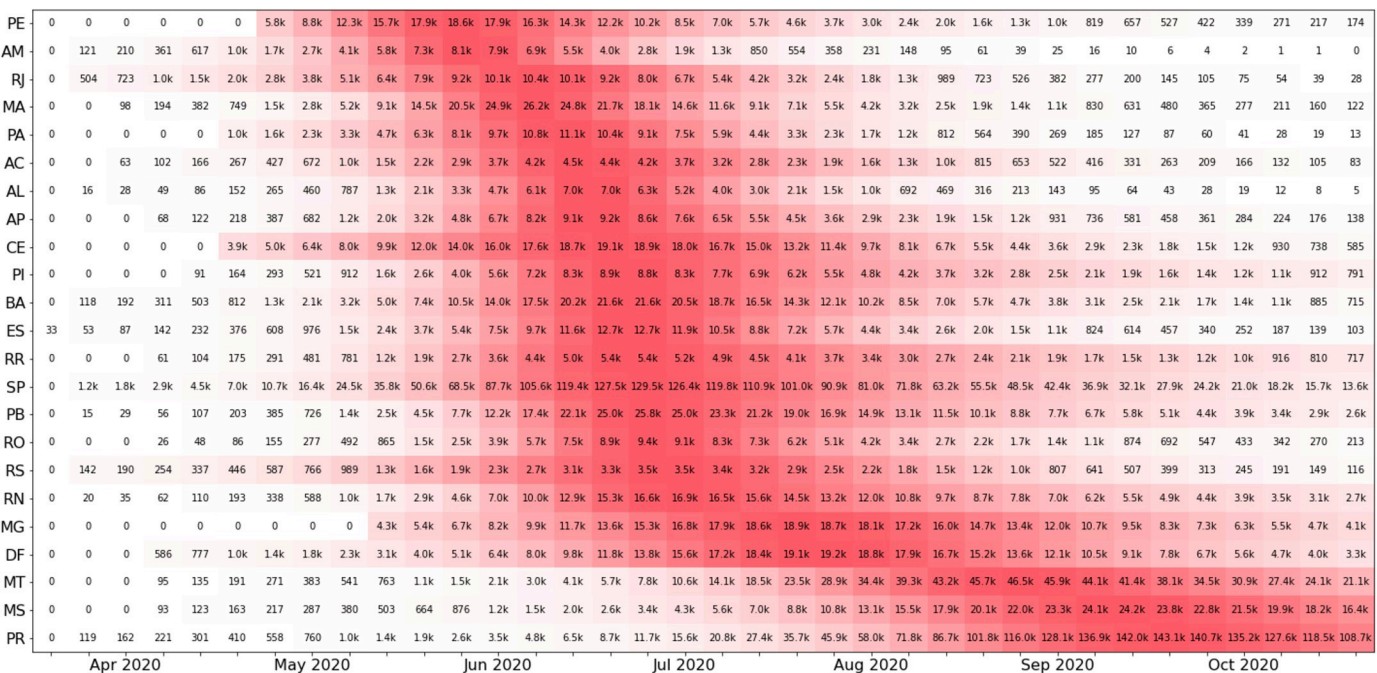

**Fig 5. Peak evolution of the epidemic in the Brazilian states illustrated in its temporal sequence of occurrence.**

suffer from the acute phase of the epidemic being Mato Grosso and Mato Grosso do Sul, which are located in the central west, and Paraná, which is located in the south. In Fig 6, the estimated parameters are presented for the ten best adjusted states for the first scenario. From the table, it is possible to see that most values of $R_0$ stay in this range, except for SC (Santa Catarina), indicating that the inclusion of the λ parameter helps with the parameter bias usually observed in direct optimisations, where it is required that λ = 1.

Considering the SIRD model for each state, it is observed that the recovery rate $D$ for individuals who accessed the health system is approximately 17 days and that the basic reproduction rate $R_0$ is approximately 2.9. Two other relevant parameters are the average mortality rate μ, which is close to 0.8%, which implies an estimated number of deaths of 128, 000 in August 2020. The rate of use of the average health system λ is on the order of 0.6% and may reach 1.0%, which implies an expectation that 2 million people will seek care in the health system. From Fig 6, it is possible to verify some other relevant features. The first is related to the comparison analysis of the parameter D. In the data, a recovered person is not a person considered to no longer be contagious, but rather a person who has been cleared by the hospital as recovered from the disease. Therefore, the state transfer dynamics, from infected to recovered, maps the time in which a person needs to receive health care until they are considered no longer affected by the disease and thus cured. That is why the estimated parameters for D have an average value of 17.97, when it is well known that a person is only contagious during their first week of symptoms. From the data, it was found that COVID-19 has a consistent value for the daily death rate, which has an average value of 0.6%. Compared with the model results, the state that is most off is RJ, with an estimated value of 2.0%. Notice however that this state that has the worst value for $R^2$ in Fig 2. This particular problem was caused by a lack of rigorous data collection. The data contain many outliers, and during a long period of time, the data-set was not updated. The synthesis of the results, based on the models by state, can be analysed in Fig 7.

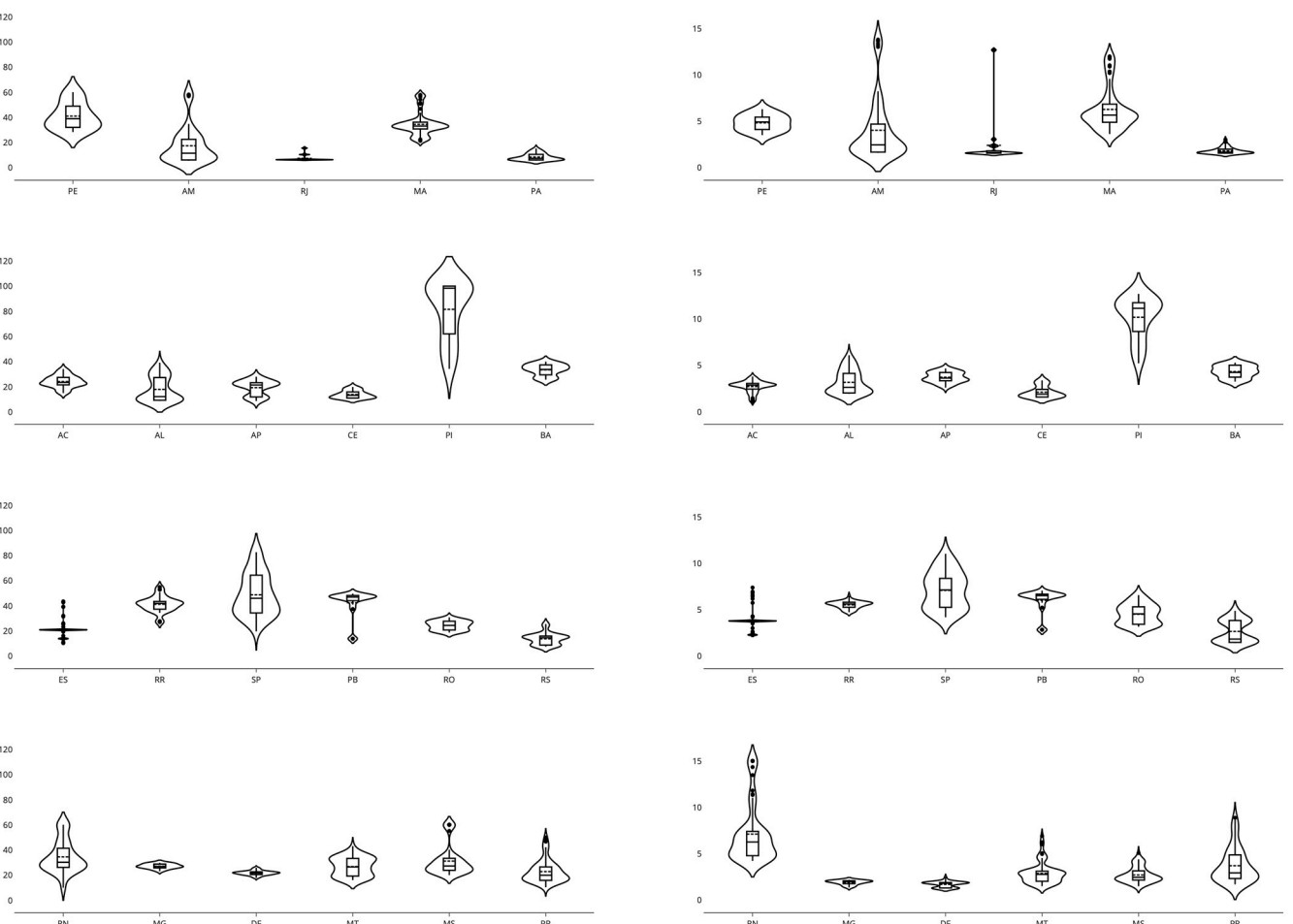

**Fig 6. Distributions of the estimated parameters $D$, $R_0$, $\mu$ and $\lambda$, for all Brazilian states based on the state that presented the first peak.** (a) Distribution of the estimated recovery rate in days: $-D - \bar{X}_D = 17.97 \pm \sigma_D = 3.41$. (b) Distribution of the estimated basic reproductive rate: $R_0 - \bar{X}_{R_0} = 2.9 \pm \sigma_{R_0} = 0.9$.

## Space-time analysis

To calculate the $R_t$ values for each Brazilian state, we used the predicted number of infected individuals, considering a window of $w$ days. The infection series can be predicted from the likelihood function considering a Poisson process [21, 22]. This procedure can be applied to both raw data and data predicted by the model. The results are illustrated in Fig 8 using a window $w = 5$ days displaced by 1 day, normalised by the z-score method. A second method, based on the extraction of the $\beta$ transmission rate, was applied for each state, resulting in a third estimate for control of the epidemic [23]. The results are illustrated in Fig 8. This last method, although probably affected by the fluctuations in the updated data in addition to social isolation factors, is shown to be correlated with the other two. The estimate of the $R_t$ rate from the model works as a filter when compared to the rate obtained from the data window, where the estimate tends to the mean the data and seems suitable for use in the prediction of contagion behavior.

As noted in Figs 1 and 8, there is a rapid recovery of the model from variations in the data. In the specific case, the variation was caused by the variation of the data sources and their consolidation, but it is expected that the model recovery will be equally rapid if the change comes

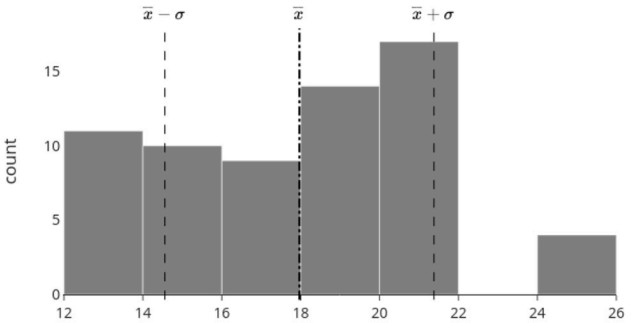

**(a) D** - removal rate in days.

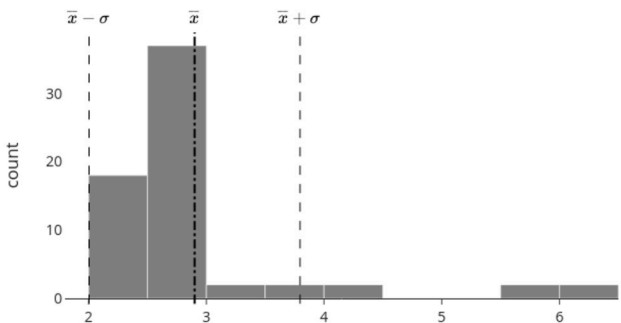

**(b)** $R_0$ - Basal reproduction rate.

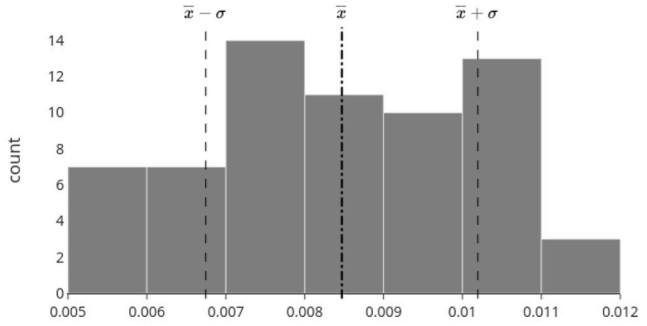

**(c)** $\mu$ - mortality rate.

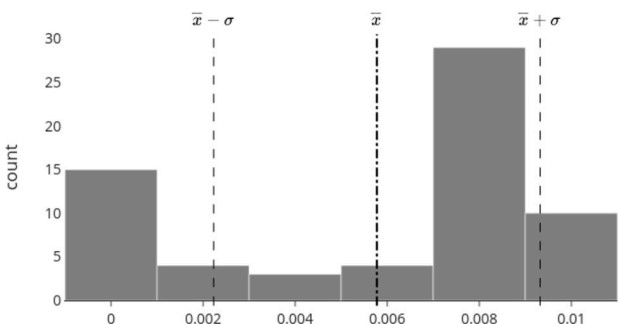

**(d)** $\lambda$ - proportionality of the population.

**Fig 7. Distributions of the estimated parameters D, $R_0$, $\mu$ e $\lambda$, accumulated since the beginning of the pandemic, for all Brazilian states.** (a) Distribution of the estimated recovery rate in days: $D - \bar{X}_D = 17.97 \pm \sigma_D = 3.41$. (b) Distribution of the estimated basic reproductive rate: $R_0 - \bar{X}_{R_0} = 2.9 \pm \sigma_{R_0} = 0.9$. (c) Distribution of the estimated mortality rate: $\mu - \bar{X}_\mu = 0.8\% \pm \sigma_\mu = 0.2\%$. (d) Distribution of the proportionality rate of the estimated population: $\lambda - \bar{X}_\lambda = 0.6\% \pm \sigma_\lambda = 0.4\%$.

from real cases. In this sense, the rates estimated from data and from the model can be used in a combined way for decision making, since they are interpreted on a 1- or 2-week horizon to observe their effects.

When we analyse the temporal and geographical progression of the virus, representing the effective reproductive index $R_t$ for each moment and each state, as shown in Fig 9, a considerable portion of the states are still observed to have indices greater than 1. Other states show a decline but are in the opening process. According to the graphs, it is visible that the epidemic started in the northern and southeastern regions. Then it spread progressively throughout the country but did not progress equally in each state. The southeastern region reaches its peak weeks later than the northern region. Compared to the northern region, the southeastern region maintains high $R_t$ values at present, showing less control of the epidemic. Note that at the moment, there are still three states that have $R_t$ values greater than 1.0, i.e., the epidemic is still growing. An example of this behaviour is the state of Paraná. As previously mentioned, according to state models, Brazil will experience a second smaller peak in September 2020.

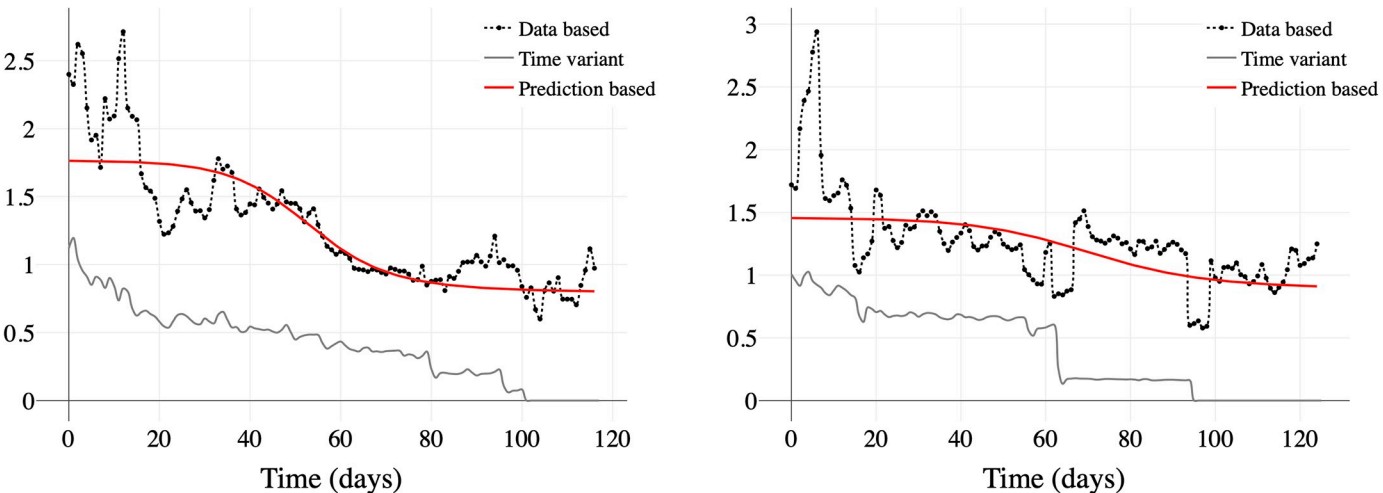

**Fig 8. Comparison of standard scaled $R_t$ from different estimate algorithms.** (a) Maranhão estimated R(t). (b) São Paulo estimated R(t).

**Fig 9. Estimated values of $R_t$ for each state during the epidemic period in Brazil.** Starting in March 2020 (upper left corner) and concluding in September 2020 (lower right corner). The maps were generated in python using the Plotly library [24].

This second peak has an amplitude mostly determined by infection numbers from the state of Paraná, which has the potential to reach values equivalent to those of São Paulo (predominant state in the amplitude of the first peak).

## Conclusion and future directions

In conclusion, our modified SIRD model allowed the estimation of the COVID-19 epidemic model for the whole Brazil and may be used in other very large countries, such as the USA, India and Russia. The results obtained and observations during the training and tuning process are compatible with other works [25]. When predicting the future of the pandemic in those countries, it is important that local variation in epidemic stage is accounted in the model to provide accurate results. The use of the composite model to understand the epidemic in Brazil allowed for a more realistic modeling, regarding the predictions of the use of the health system, as well as the average control parameters of the epidemic. We also found that COVID-19 peaked in Brazilian states during periods in which the peak of respiratory diseases also used to occur. At the time of the writing of this paper, June 2020, the values of $R_0$ and $R_t$ higher than 1 found for Brazilian States and the high values predicted until the last quarter of 2020 suggests that non-pharmacological measures would be needed for months, what turned out to be true. Another aspect that the model brought as evidence of its predictive capacity was the stable level between the months of June and October, with the beginning of a decline in cases after a considerable period of stable number of cases.

As a management element for a country of continental dimensions such as Brazil, the model proved to be effective in providing information to support decision making in the public and private spheres. However, its predictive capacity can be expanded considering aspects of closure and opening of economically active segments of society, such as the proposal studied in [26]. One feature of the model is that it needs to be updated as new data become available daily to improve its estimates. Limitations to be addressed refer to the characterisation of the initial state of the epidemic, prediction of the possibility of new waves, and the inclusion of vaccination processes. In order to improve its capacity, increasing the complexity, we can analyse the impact of the quarantine as in the model tuned in Dynamic-Susceptible-Exposed-Infective-Quarantined (D-SEIQ) [27].

As part of the dissemination strategy for this work, we created a repository that allows interested parties to access all the code described in python as well as the preliminary tuning results. We intend to improve communication with the inclusion of information about vaccination, risk analysis as proposed in [28, 29], and also modifications of the model in the case of incorporating the latency aspect in the transfer, which has gained greater understanding in the light of new observations [30].

## Author Contributions

**Conceptualization:** V. C. Parro.

**Formal analysis:** V. C. Parro, T. N. Toporcov.

**Funding acquisition:** V. C. Parro.

**Investigation:** V. C. Parro, F. B. Ipólito.

**Methodology:** V. C. Parro, F. Pait, T. N. Toporcov.

**Software:** M. L. M. Lafetá, F. B. Ipólito.

**Supervision:** V. C. Parro.

**Validation:** M. L. M. Lafetá, F. Pait, F. B. Ipólito.

**Visualization:** M. L. M. Lafetá, F. B. Ipólito.

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
