## [Decision Letter · Decision Letter 0]

8 Dec 2020

PONE-D-20-34349

Predicting COVID-19 in very large countries: the case of Brazil

PLOS ONE

Dear Dr. Parro,

Thank you for submitting your manuscript to PLOS ONE. After careful consideration, we feel that it has merit but does not fully meet PLOS ONE’s publication criteria as it currently stands. Therefore, we invite you to submit a revised version of the manuscript that addresses the points raised during the review process.

We look forward to receiving your revised manuscript.

Kind regards,

Seyedali Mirjalili

Academic Editor

PLOS ONE

Journal Requirements:

2.Thank you for stating the following in the Acknowledgments Section of your manuscript:

"The authors would like to thank the Instituto Maua de Tecnologia for funding this work."

3. We noted in your submission details that a portion of your manuscript may have been presented or published elsewhere. (preprint on Research square.)

4. We note that Figure 9 in your submission contain map images which may be copyrighted. All PLOS content is published under the Creative Commons Attribution License (CC BY 4.0), which means that the manuscript, images, and Supporting Information files will be freely available online, and any third party is permitted to access, download, copy, distribute, and use these materials in any way, even commercially, with proper attribution. For these reasons, we cannot publish previously copyrighted maps or satellite images created using proprietary data, such as Google software (Google Maps, Street View, and Earth). For more information, see our copyright guidelines: http://journals.plos.org/plosone/s/licenses-and-copyright.

(1) You may seek permission from the original copyright holder of Figure 9 to publish the content specifically under the CC BY 4.0 license. 

(2) f you are unable to obtain permission from the original copyright holder to publish these figures under the CC BY 4.0 license or if the copyright holder’s requirements are incompatible with the CC BY 4.0 license, please either i) remove the figure or ii) supply a replacement figure that complies with the CC BY 4.0 license. Please check copyright information on all replacement figures and update the figure caption with source information. If applicable, please specify in the figure caption text when a figure is similar but not identical to the original image and is therefore for illustrative purposes only.

Reviewers' comments:

Reviewer's Responses to Questions

**Comments to the Author**

1. Is the manuscript technically sound, and do the data support the conclusions?

Reviewer #1: Yes

Reviewer #2: Yes

2. Has the statistical analysis been performed appropriately and rigorously? 

Reviewer #1: Yes

Reviewer #2: Yes

3. Have the authors made all data underlying the findings in their manuscript fully available?

Reviewer #1: Yes

Reviewer #2: Yes

4. Is the manuscript presented in an intelligible fashion and written in standard English?

Reviewer #1: Yes

Reviewer #2: Yes

5. Review Comments to the Author

Reviewer #1: The authors presented a logical modified version of numerical model of susceptible, Infected, Removed and dead (SIRD) model to estimate health system use for COVID-19 cases. This realistic machine learning (ML) approach is not only suitable for Brazil but also for other large countries like USA, India and Russia. Moreover, the developed system is accessible on an open platform and the best part is that it is possible to receive a synthesis report, synchronized model and python source code once a standard CSV file is submitted. Optimization algorithm using heuristic search is also introduced into the learning algorithm.

Overall, great work has been presented by the authors. The beauty of the proposed work is its simplicity that is a boost for the use by public agents in management and communication. When compared with other models, it is simpler to tune the model’s parameters. It allows all relevant parameters for public management and decision making such as its main parameters, infection rate, removal rate, basic reproduction rate R_0 and effective reproduction rate R_t to do a comparative assessment. The authors successfully validated the model on 26 Brazilian states.

Comments:

The topic is interesting however the study has few flaws and requires minor revisions in order to improve the study in the following ways:

Graphical/Tabular performance analysis of the proposed work with other existing ML models

Inclusion of limitation of the model

Conclusion may be further elaborated (page 18)

Check on the repetition of the same line twice where ∆t and P is mentioned (2nd paragraph, page 9)

Reviewer #2: The work presents a new proposal for estimating health system use for COVID-19 cases for the whole Brazil. The paper seems good, well-described and structured. However, the comments and the requirements given below should be addressed before accepting the paper

1 There are some grammar and spelling mistakes in the paper, the language quality should be enhanced.

2 In section 2 on page 3 (Section the model), the sentence (where ∆t is the sample time of the data-sets, and P is the total population that should be considered for each case study) is repeated. The repeated sentence must be removed.

3 All symbols in equations should be clarified.

4 The paper needs to add a review section (Related works) and include some well-established works on predicting COVID-19 that done until now.

5 Add some suggestions for further works and change the title of "conclusion" section to "conclusion and future directions".

6. PLOS authors have the option to publish the peer review history of their article (what does this mean?). If published, this will include your full peer review and any attached files.

Reviewer #1: No

Reviewer #2: **Yes: **Yassine Meraihi

---

## [Author Response · Author response to Decision Letter 0]

25 Feb 2021

All reviewers questions were discussed in the document Response to reviewers .pdf

---

## [Decision Letter · Decision Letter 1]

11 Apr 2021

PONE-D-20-34349R1

Predicting COVID-19 in very large countries: the case of Brazil

PLOS ONE

Dear Dr. Parro,

Thank you for submitting your manuscript to PLOS ONE. After careful consideration, we feel that it has merit but does not fully meet PLOS ONE’s publication criteria as it currently stands. Therefore, we invite you to submit a revised version of the manuscript that addresses the points raised during the review process.

Based on the reviewer's and my own suggestions, I recommend major revisions for this paper.

We look forward to receiving your revised manuscript.

Kind regards,

Thippa Reddy Gadekallu

Academic Editor

PLOS ONE

Journal Requirements:

Reviewers' comments:

Reviewer's Responses to Questions

**Comments to the Author**

1. If the authors have adequately addressed your comments raised in a previous round of review and you feel that this manuscript is now acceptable for publication, you may indicate that here to bypass the “Comments to the Author” section, enter your conflict of interest statement in the “Confidential to Editor” section, and submit your "Accept" recommendation.

Reviewer #2: All comments have been addressed

Reviewer #3: (No Response)

2. Is the manuscript technically sound, and do the data support the conclusions?

Reviewer #2: Yes

Reviewer #3: Yes

3. Has the statistical analysis been performed appropriately and rigorously? 

Reviewer #2: Yes

Reviewer #3: Yes

4. Have the authors made all data underlying the findings in their manuscript fully available?

Reviewer #2: Yes

Reviewer #3: Yes

5. Is the manuscript presented in an intelligible fashion and written in standard English?

Reviewer #2: Yes

Reviewer #3: Yes

6. Review Comments to the Author

Reviewer #2: The work presents a new proposal for estimating health system use for COVID-19 cases for the whole Brazil. The paper seems good, well-described and structured. I accept the paper

Reviewer #3: 1. The English language in the paper is loose in some instances. The paper needs a thorough proofread.

2. There are several long sentences in the paper. The authors can break them down into smaller sentences.

3. List out the main contributions of the paper.

4. Some of the recent works such as the following can be discussed in the paper: "Deep learning and medical image processing for coronavirus (COVID-19) pandemic: A survey, An Incentive Based Approach for COVID-19 planning using Blockchain Technology".

5. Discuss about the limitations of the present work.

7. PLOS authors have the option to publish the peer review history of their article (what does this mean?). If published, this will include your full peer review and any attached files.

Reviewer #2: **Yes: **Yassine Meraihi

Reviewer #3: No

---

## [Author Response · Author response to Decision Letter 1]

14 May 2021

All answers and comments were included at the Response to reviewers file.

---

## [Decision Letter · Decision Letter 2]

31 May 2021

Predicting COVID-19 in very large countries: the case of Brazil

PONE-D-20-34349R2

Dear Dr. Parro,

We’re pleased to inform you that your manuscript has been judged scientifically suitable for publication and will be formally accepted for publication once it meets all outstanding technical requirements.

Kind regards,

Thippa Reddy Gadekallu

Academic Editor

PLOS ONE

Additional Editor Comments (optional):

Reviewers' comments:

Reviewer's Responses to Questions

**Comments to the Author**

1. If the authors have adequately addressed your comments raised in a previous round of review and you feel that this manuscript is now acceptable for publication, you may indicate that here to bypass the “Comments to the Author” section, enter your conflict of interest statement in the “Confidential to Editor” section, and submit your "Accept" recommendation.

Reviewer #3: All comments have been addressed

2. Is the manuscript technically sound, and do the data support the conclusions?

Reviewer #3: Yes

3. Has the statistical analysis been performed appropriately and rigorously? 

Reviewer #3: Yes

4. Have the authors made all data underlying the findings in their manuscript fully available?

Reviewer #3: Yes

5. Is the manuscript presented in an intelligible fashion and written in standard English?

Reviewer #3: Yes

6. Review Comments to the Author

Reviewer #3: The authors have addressed all the comments and incorporated all my suggestions. The paper can be accepted for publication.

7. PLOS authors have the option to publish the peer review history of their article (what does this mean?). If published, this will include your full peer review and any attached files.

Reviewer #3: No

---

## [Editor Report · Acceptance letter]

23 Jun 2021

PONE-D-20-34349R2 

Predicting COVID-19 in very large countries: the case of Brazil 

Dear Dr. Parro:

I'm pleased to inform you that your manuscript has been deemed suitable for publication in PLOS ONE. Congratulations! Your manuscript is now with our production department. 

Kind regards, 

on behalf of

Dr. Thippa Reddy Gadekallu 

Academic Editor

PLOS ONE